# Correlation between Maternal Weight Gain in Each Trimester and Fetal Growth According to Pre-Pregnancy Maternal Body Mass Index in Twin Pregnancies

**DOI:** 10.3390/medicina58091209

**Published:** 2022-09-02

**Authors:** Mi Ju Kim, Hyun Mi Kim, Hyun-Hwa Cha, Won Joon Seong

**Affiliations:** Department of Obstetrics and Gynecology, Kyungpook National University Hospital, School of Medicine, Kyungpook National University, Daegu 41944, Korea

**Keywords:** twin pregnancy, maternal weight gain, fetal growth, maternal complications, neonatal outcomes

## Abstract

*Background and**Objectives*: This study aimed to determine the correlation between maternal weight gain in each trimester and fetal growth according to pre-pregnancy maternal body mass index in twin pregnancies. *Materials and Methods*: We conducted a retrospective review of the medical records of 500 twin pregnancies delivered at 28 weeks’ gestation or greater at a single tertiary center between January 2011 and December 2020. We measured the height, pre-pregnant body weight, and maternal body weight of women with twin pregnancies and evaluated the relationship between the maternal weight gain at each trimester and fetal growth restriction according to pre-pregnancy body mass index. *Results*: The overweight pregnant women were older than the normal or underweight pregnant women, and the risk of gestational diabetes was higher. The underweight pregnant women were younger, and the incidence of preterm labor and short cervical length during pregnancy was higher in the younger group. In normal weight pregnant women, newborn babies’ weight was heavier when their mothers gained weight, especially when they gained weight in the second trimester. Mothers’ weight gain in the first trimester was not a significant factor to predict fetal growth. The most predictive single factor for the prediction of small neonates was weight gain during 24–28 and 15–18 weeks, and the cutoff value was 6.2 kg (area under the curve 0.592, *p* < 0.001). *Conclusions*: In twin pregnancy, regardless of the pre-pregnant body mass index, maternal weight gain affected fetal growth. Furthermore, weight gain in the second trimester of pregnancy is considered a powerful indicator of fetal growth, especially in normal weight pregnancies.

## 1. Introduction

Twin pregnancies account for about 1–3% of all pregnancies. The incidence of twin pregnancies has been increasing due to advanced maternal age and the development of assisted reproductive techniques (ARTs) [1,2]. Twin pregnancies involve complications, including twin-specific problems. Compared with singleton pregnancy, twin pregnancy not only increases the possibilities of obstetric complications, such as preterm labor, preeclampsia, gestational diabetes, and operative delivery [3,4], but also many neonatal problems, such as miscarriage, fetal death in utero, preterm birth, and fetal growth restriction [5,6,7]. Many attempts have been made to improve the obstetric or neonatal prognosis in twin pregnancy.

In singleton pregnancy, some studies have shown that proper weight gain during pregnancy is associated with obstetric prognosis. Many researchers have suggested criteria for appropriate weight gain during pregnancy based on the maternal pre-pregnancy weight [8,9].

In twin pregnancy, proper maternal weight gain during pregnancy is associated with pregnancy-related complications and neonatal outcomes, as in singleton pregnancy [10,11,12]. If the mother is tall, and the maternal weight gain during pregnancy is appropriate, maternal weight gain can predict fetal growth. The Institute of Medicine and the National Research Council in 1990 and 2009, respectively, presented criteria for adequate weight gain in twin pregnancy [3,13]. However, determining the correlation between maternal weight gain and gestational age of pregnancy was difficult, as it focused on the total weight gain during the entire pregnancy period. Moreover, unlike in singleton pregnancies, twin pregnancy mothers have a higher likelihood of weight gain and pregnancy-related complications, such as fetal growth restriction, preeclampsia, or gestational diabetes [3,4]. Furthermore, defining proper weight gain is difficult without considering race, socioeconomic status, maternal health, and maternal height. Twin pregnancy particularly often results in early delivery due to early labor. The twin pregnancy itself increases the risk of newborns who are small for gestational age (SGA). SGA can increase several morbidities, such as neonatal intensive care unit (NICU) admission, respiratory complications, and jaundice. Therefore, presenting appropriate weight gain values during pregnancy according to the trimester of pregnancy will help prevent and predict SGA.

This study sought to investigate the relationship between fetal growth and maternal weight gain in different trimesters of twin pregnancy. Based on the weight of the neonates, the patients were divided into two groups: (1) AGA (adequate for gestational age) group (AGA–AGA) and (2) SGA group (AGA–SGA or SGA–SGA). Maternal weight and weight gain were compared between the groups. In addition, the patients were divided into three groups according to pre-pregnant BMI (body mass index): underweight, normal, and overweight. The relationship between maternal weight gain and fetal growth was assessed in each group. We aimed to find out whether the amount of maternal weight gain during pregnancy could be a significant prognostic factor of SGA in twin pregnancy.

## 2. Materials and Methods

We conducted a retrospective medical record review of 500 Korean patients with twin pregnancies who delivered at 28 weeks’ gestation or greater at a single center (Chilgok Kyungpook National University Hospital) in Daegu, South Korea, between January 2011 and December 2020. Forty-two women were excluded because of major fetal anomalies, chromosomal abnormalities, and fetal death in utero. Four women with monochorionic–monoamniotic twins were also excluded. When the chorionicity in the medical records and placental pathologic findings after delivery differed, the pathologic results were followed. Twelve cases of twin-to-twin transfusion syndromes were also excluded. Overall, 58 pregnancies were excluded based on the exclusion criteria, and 442 twin pregnancies were reviewed. Maternal height, pre-pregnant body weight, and body weight were measured. Of these, we analyzed the maternal body weights between 15 and 18 weeks of gestation, 20 and 24 weeks of gestation, and 24 and 28 weeks of gestation, as well as weight at delivery. We also evaluated the differences in maternal body weight at each period and determined the weight gain during pregnancy. Ninety-nine pregnancies that did not have weight measurement data at each period were excluded. Thus, the final number of pregnancies included in the analyses was 343.

BMI was measured using maternal height and weight before pregnancy. The groups with BMI values of less than 18.5 were classified as underweight, the groups with BMI values of 25 or higher were classified as overweight, and the groups with BMI values between 18.5 and 25 were classified as normal weight. The obstetric and neonatal outcomes of each group were compared.

Maternal characteristics, such as age at delivery, height, parity, the use of ARTs, chorionicity, and pregnancy-related complications, such as preeclampsia, placenta previa, overt diabetes, gestational diabetes (GDM), threatened preterm labor, short cervical length, premature membrane rupture (PROM), and postpartum bleeding, were compared. Preeclampsia was defined as having a blood pressure of 140/90 mmHg or more and proteinuria after 20 weeks of gestation in accordance with the criteria of the International Society for the Study of Hypertension in Pregnancy. For GDM, a 50 g glucose tolerance test was performed as screening between 24 and 28 weeks of gestation, and a 100 g glucose tolerance test was performed when the value was over 140 mg/dL. GDM was diagnosed in accordance with the Carpenter–Coustan criteria. Placenta previa was defined by evaluating the location of the placenta through ultrasound immediately before delivery. Threatened preterm labor was defined as inpatient conservative treatment before 37 weeks of gestation due to regular uterine contractions and a short cervical length of less than 2.5 cm. Postpartum bleeding was defined as massive bleeding of more than 1 L or requiring transfusion due to anemia after delivery. Deliveries scheduled on a date were classified as elective, deliveries due to labor pain were classified as spontaneous, and emergency deliveries due to preeclampsia or poor fetal condition were classified as iatrogenic.

The neonatal outcomes, including sex, gestational age at delivery, neonatal birth weight, the rate of SGA, twin weight discordance, 1 min Apgar score, 5 min Apgar score, and NICU admission, were analyzed. Weight below the 10th percentile corrected for gestational age and sex was defined as SGA, and more than that was defined as AGA [14]. Twin weight discordance was determined using neonatal birth weight multiplied by 100 after dividing the difference between the weights of the two neonates by the weight of the large baby, as defined in previous studies. If the value obtained was more than 20%, the twins were classified as discordant twins, and if the value was less than 20%, the twins were classified as concordant [15]. Each of the normal, underweight, and overweight groups were further divided based on the neonatal weights: AGA–AGA group and AGA–SGA or SGA–SGA group. Maternal weight and weight gain during pregnancy were also compared. This study was approved by the Institutional Review Board (IRB) of Chilgok Kyungpook National University Hospital (IRB no. KNUCH 2021-04-037). Informed consent was not obtained from the study participants due to the retrospective nature of the study in reviewing medical records.

### Statistical Analyses

All data were analyzed using the SPSS software (version 26.0; SPSS Inc., Chicago, IL, USA) and R version 4.0.0 (Vienna, Austria; www.r-project.org/ accessed on 24 April 2020). The data on the AGA, AGA–SGA, and SGA–SGA groups were compared using the Mann–Whitney U test for continuous numerical data, whereas the χ^2^ test was used for binary categorical data. Data were presented as mean ± standard deviation (SD) for continuous variables with normal distributions and number (percentage) for binary categorical data. A *p* value < 0.05 was considered statistically significant.

## 3. Results

This study included 343 twin pregnancies and 686 neonates. The patients were divided into underweight, normal weight, and overweight, which comprised 9.6%, 71.1%, and 19.2% of the study population, respectively.

Table 1 compares the maternal characteristics and pregnancy-related complications according to pre-pregnant BMI. Compared with the normal weight group, underweight twin pregnancies were significantly younger, with a mean ± SD age at delivery of 30.24 ± 5.13 years (vs. 32.28 ± 4.13 years, *p* = 0.003). Their pre-pregnancy weight was significantly lower, and their weight gain was also significantly lower than in the normal weight group (13.04 ± 5.35 kg vs. 14.72 ± 7.13 kg, *p* = 0.024). However, there was no statistically significant difference in parity, maternal height, the use of ARTs, and chorionicity. The incidence of preeclampsia in underweight pregnancies was 3.03%, which was significantly lower than that in the normal BMI with 13.11% (*p* = 0.030). However, the frequency of preterm labor was significantly higher in the underweight group (81.82% vs. 58.20%, *p* < 0.001). In the underweight group, emergency delivery due to reasons such as spontaneous labor or rupture of membrane (ROM) was significantly higher than that in the normal weight group (60.61% vs. 45.68%, *p* = 0.035). In the overweight group, age at delivery was 33.44 ± 3.95 years old and was significantly higher than that in the normal weight group (*p* = 0.004). Weight gain during pregnancy in the overweight group was 11.53 ± 5.49 kg, which was significantly lower than that in the normal weight group (14.72 ± 7.13 kg, *p* < 0.001). There was no significant difference in the use of ART (54.55% vs. 55.14%, *p* = 0.880), but the frequency of dichorionic twins tended to be higher in the overweight group (84.85% vs. 75.82%, *p* = 0.036). In the overweight group, the frequencies of chronic hypertension (3.03% vs. 0.00%, *p* = 0.001) and GDM were higher (27.27% vs. 10.25%, *p* < 0.001), but the frequency of preeclampsia was lower compared with the normal weight group (6.06% vs. 13.11%, *p* = 0.037). The cause of delivery did not differ significantly between the overweight and normal weight groups.

Table 2 compares the neonatal outcomes according to pre-pregnant BMI. Delivery in the underweight group was earlier compared with the normal weight group (34.08 ± 2.90 weeks vs. 34.93 ± 2.89 weeks, *p* = 0.025), and neonatal birth weight was lower (1939.02 ± 228.46 g vs. 2193.02 ± 521.92 g, *p* < 0.001). However, the incidence of SGA did not vary significantly between both groups (19.70% vs. 19.06%, *p* = 1.000). There was no significant difference in twin weight discordance, neonatal outcomes, such as Apgar score at 1 and 5 min, NICU admission, the requirement for oxygen, and intubation. Neonatal birthweight percentile was higher in the overweight group compared with the normal weight group, (30.14 ± 21.23 vs. 36.61 ± 22.46, *p* = 0.002), but no significant difference was observed in gestational age at delivery, sex, neonatal birthweight, and incidence of SGA. Furthermore, no statistical difference was observed in the neonatal outcomes, such as discordancy, Apgar score at 1 and 5 min, NICU admission, the requirement for oxygen, and intubation.

Table 3 compares the obstetrical complications, neonatal prognosis, and maternal weight gain between the AGA–AGA group and AGA–SGA or SGA–SGA group, regardless of BMI based on maternal pre-pregnancy weight. In the AGA–SGA or SGA–SGA group, the maternal age was significantly higher (33.67 ± 3.97 years vs. 31.67 ± 4.27 years, *p* < 0.001), and there were significantly more nulliparous pregnancies in these groups than in AGA–AGA groups (76.15% vs. 65.38%, *p* = 0.001), and significantly more conceptions through ART (65.14% vs. 48.93%, *p* < 0.001) compared with those in the AGA group. There was no significant difference in the frequency of GDM, but the tendency of preeclampsia was significantly higher in the SGA group (16.51% vs. 8.12%, *p* = 0.002). Compared with the AGA–SGA or SGA–SGA group, the risk of threatened preterm labor significantly increased (69.23% vs. 41.28%, *p* < 0.001), the presence of short cervical lengths below 2.5 cm was significantly higher (53.85% vs. 29.47%, *p* < 0.001), and the rate of PROMs was also significantly higher (29.06% vs. 16.51%, *p* = 0.001) in the AGA–AGA group. There was a higher frequency of spontaneous delivery than elective or iatrogenic (*p* < 0.001) in the SGA group. Compared with the AGA–AGA group, the SGA–SGA or AGA–SGA group had a higher gestational age at delivery (35.81 ± 2.24 weeks vs. 34.37 ± 2.99 weeks, *p* < 0.001), a lower birth weight (2117.34 ± 474.37 g vs. 2206.85 ± 558.34 g, *p* = 0.030), and a smaller placenta. The frequency of NICU hospitalization was higher in the SGA group (77.06% vs. 68.45%, *p* = 0.025), but no significant difference was observed in other neonatal outcomes in both groups. In the SGA group, the women were shorter than those in the AGA group (159.85 ± 4.72 vs. 161.40 ± 5.10 cm, *p* < 0.001), had a lower pre-pregnancy weight (55.83 ± 9.41 vs. 58.52 ± 11.07 kg, *p* = 0.001), and had a lower weight at delivery (69.65 ± 9.15 vs. 72.26 ± 12.30 kg, *p* = 0.002). Weight measurements at 15–18 weeks and 24–28 weeks of gestation were also lower in the SGA group.

Table 4 compares the maternal weight and weight gain according to gestational age between AGA–AGA and AGA–SGA or SGA–SGA in the normal weight group. In the SGA sub-group, the maternal height was lower (160.14 ± 4.25 vs. 161.32 ± 5.22 cm, *p* = 0.008), but no statistical difference was found between the two groups in pre-pregnancy weight and weight at delivery compared with the AGA group. However, the weight at 15–18 weeks (56.19 ± 5.64 vs. 58.42 ± 6.37 kg, *p* = 0.004) and 24–28 weeks of gestation (63.27 ± 5.23 vs. 65.54 ± 6.93 kg, *p* = 0.003) tended to be lower. Comparison according to trimesters showed that the SGA group had a significantly smaller weight gain in the second trimester of pregnancy compared with the AGA group (6.50 ± 2.04 vs. 7.21 ± 2.87 kg, *p* = 00043). On the other hand, weight gain in the third trimester of pregnancy was higher in the SGA group (6.27 ± 2.96 vs. 5.00 ± 3.67 kg, *p* = 0.003).

Table 5 compares the maternal weight and weight gain according to gestational age between AGA–AGA and AGA–SGA or SGA–SGA in the overweight group. In the SGA group, gestational age at delivery was higher (35.52 ± 1.97 vs. 34.54 ± 2.82 weeks, *p* = 0.032) and the maternal height was smaller (158.38 ± 6.25 vs. 160.97 ± 5.09 cm, *p* = 0.020) compared with the AGA group. No significant difference in pre-pregnancy weight was observed between the two groups. However, the weights at 15–18 weeks (75.84 ± 8.49 vs. 70.43 ± 5.79 kg, *p* = 0.001) and 24–28 weeks of gestation (82.85 ± 9.85 vs. 75.61 ± 6.07 kg, *p* = 0.001) and at delivery (86.42 ± 11.06 vs. 80.29 ± 11.77 kg, *p* = 0.008) were lower. Comparison according to trimesters showed no significant difference in weight gain between the SGA and AGA group.

Table 6 compares the maternal weight and weight gain according to gestational age between the AGA–AGA and AGA–SGA or SGA–SGA in the underweight group. The SGA group had a higher gestational age at delivery (35.11 ± 2.24 vs. 33.49 ± 3.09 weeks, *p* = 0.028), and smaller maternal height (159.67 ± 5.30 vs. 162.48 ± 3.82 cm, *p* = 0.015) compared with the AGA group. No statistical difference was observed between the two groups in pre-pregnancy weight. The weight at 15–18 weeks of gestation was lower (46.78 ± 4.35 vs. 50.17 ± 4.03 kg, *p* = 0.022). However, no significant difference was observed in the weights at 20–24 weeks of gestation, at 24–28 weeks of gestation, and at delivery. Moreover, there was no significant difference in weight gain according to trimesters between the AGA and SGA groups.

Figure 1 shows the area under the curve (AUC) of the predictive values for maternal weight and weight gain during pregnancy for SGA in twin pregnancy. The single predictive factor for SGA was weight gain during 24–28 weeks and 15–18 weeks, i.e., during the second trimester of pregnancy, and the cutoff value was 6.2 kg (AUC = 0.592, *p* < 0.001). The combined factors for the prediction of SGA in twin pregnancy were weight gain between 24–28 and 15–18 weeks and weight gain between 20–24 and 15–18 weeks, with an AUC of 0.585 at cutoff values of 6.8 and 4 kg, respectively (*p* < 0.001).

## 4. Discussion

In this study, the patients in the overweight group were older, had smaller maternal weight gains during pregnancy, and had a higher risk of GDM than those in the normal or underweight group. However, there was no significant difference in the incidence of preterm labor or abnormal placentation. In singleton pregnancy, maternal overweight or obesity is known to increase pregnancy-related complications [16,17], but this needs to be investigated further in twins. Some studies reported that twin pregnancy, unlike singleton pregnancy, was not associated with pre-pregnancy overweight and GDM [18], but other studies have indicated that a higher BMI before pregnancy resulted in a higher rate of GDM [19]. This finding is consistent with that of the present study.

In our study, the patients in the underweight group were younger at delivery than the normal or overweight group and had a smaller weight gain during pregnancy. Furthermore, the rate of preterm labor and short cervical length during pregnancy was higher in the underweight group; thus, the possibility of emergency delivery was higher. According to Lilly et al., if weight gain by underweight mothers during pregnancy reaches the recommended value for normal BMI mothers, the frequency of preterm labor and pregnancy-related complications does not increase [20]. We found in a study of Asians that underweight women had relatively smaller weight gains, presumably due to cultural and ethnic difference from the westerners who have been the main subjects of studies thus far.

Comparing the birthweight with weight percentile values corrected for gestational age at delivery and sex showed that the neonatal size decreased in the order of overweight, normal weight, and underweight groups. However, no significant difference was observed in the incidence of SGA and in the short-term prognosis of neonates among the groups. However, this study confirmed that the probability of an SGA baby being born to an underweight woman is about 20%, which is significantly lower than the number reported in other studies (50–60%) [21]. This may be due to the differences in socioeconomic status, nutritional support, and lifestyle, including smoking or alcohol.

In twin pregnancy, if at least one twin has a fetal growth restriction of less than 10%, the average maternal age and the incidence of preeclampsia tended to be higher, which may negatively affect fetal growth [22,23]. The AGA–AGA group had more cases of threatened preterm labor, short cervical length, and premature membrane rupture than the SGA group, all of which resulted in preterm birth. This may be because the heavier the fetus, the larger the placenta and the amount of amniotic fluid, resulting in the overextension of the uterus and subsequent early labor. However, no significant difference was observed in the short-term outcomes of newborns between the two groups except for NICU hospitalization. Furthermore, the taller the pregnant woman is, the heavier the maternal weight before pregnancy. Conversely, the greater the weight gain by trimester of pregnancy, the more likely the neonate is AGA. This is consistent with the finding of another study indicating that when normal weight pregnant women achieve the recommended weight gain during pregnancy, prognosis related to several pregnancy complications, such as SGA and preterm birth, is better [24,25,26]. Bodnar et al. reported that across all BMI groups, if the weight gain is too small or too much, the risk of infant death increases. Furthermore, weight gain and SGA are inversely proportional, whereas weight gain and LGA or cesarean delivery are positively proportional [27].

In the present study, in normal weight pregnant women, newborn babies’ weight was heavier when their mothers gained weight, especially when they gained weight in the second trimester. Mothers’ weight gain in the first trimester was not a significant factor to predict fetal growth. Furthermore, the reason for the higher incidence of weight gain in the third trimester in the SGA group is probably the relatively later delivery in this group. Wen et al. reported that in dichorionic twins, weight gain in early pregnancy was found to be associated with birth weight [28]. Lilly et al. reported that weight gain in the second and first trimesters of pregnancy is associated with fetal growth, whereas weight gain after 24 weeks is associated with preterm labor [20]. However, the above studies are different from our research in that these were studies on normal weight twin pregnancies. In addition, the possibility of SGA in at least one of the two fetuses was higher when the weight increase in the second trimester of pregnancy was less than 6.2 kg in the Korean population. Thus, the possibility of SGA can be predicted using weight gain in the second trimester of pregnancy, which can help improve the prognosis.

The strength of this study is that the single center from which the data were obtained had consistently evaluated pregnant women of a single ethnicity population, measured the weight of all pregnancies in a series using the same machine from the beginning of each pregnancy. Thus, a cutoff value was set to create an SGA prediction model. The limitations of this study include a small size of the study population, an unequal number of participants for each BMI group, and selection bias, considering that only a single tertiary center was evaluated.

## 5. Conclusions

In twin pregnancy, regardless of pre-pregnant BMI, maternal weight gain affects fetal growth. Particularly, weight gain in the second trimester of pregnancy can be considered a powerful indicator for fetal growth, especially in normal weight pregnancies.

## Figures and Tables

**Figure 1 medicina-58-01209-f001:**
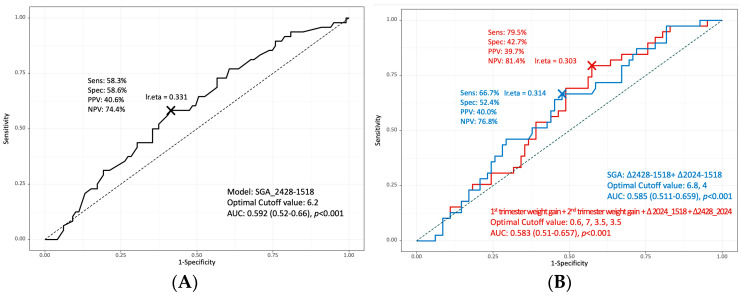
Area under the curve (AUC) of the predictive values for maternal weight and weight gain during pregnancy for the small for gestational age (SGA) group in twin pregnancy. (**A**) Among various weight gain values by pregnancy period, weight gains during 24–28 and 15–18 weeks, i.e., during the second trimester of pregnancy, were the most significant predictive factor, with an AUC = 0.592 and a cutoff value of 6.2 kg. (**B**) The weight gain value for each pregnancy trimester was simultaneously analyzed, and the AUC value that can predict the rate of SGA was confirmed. The blue line indicates weight gain between 24–28 and 15–18 weeks and between 20–24 and 15–18 weeks, with an AUC of 0.585 and cutoff values of 6.8 and 4 kg, respectively. The orange line indicates the SGA prediction using weight gain in the first trimester, weight gain in the second trimester, weight gain between 20–24 and 15–18 weeks, and weight gain between 24–28 and 20–24 weeks, with cutoff values of 0.6, 7, 3.5, and 3.5 kg, respectively (AUC = 0.583).

**Table 1 medicina-58-01209-t001:** Maternal characteristics and pregnancy-related complications according to maternal pre-pregnant BMI in twin pregnancy.

	Underweight (*n* = 33)	Normal Weight (*n* = 244)	Overweight (*n* = 66)	*p* Value	*p* Value α	*p* Value β	*p* Value γ
Maternal age, years	30.24 ± 5.13	32.28 ± 4.13	33.44 ± 3.95	<0.001 *	0.003 *	<0.001 *	0.004 *
Nulliparous, *n* (%)	23 (69.70%)	172 (70.49%)	41 (62.12%)	0.018 *	0.237	0.040 *	0.022 *
Height, cm	161.45 ± 4.58	160.97 ± 4.95	160.43 ± 5.51	0.153	0.450	0.194	0.279
Weight gain during pregnancy, kg	13.04 ± 5.35	14.72 ± 7.13	11.53 ± 5.49	0.006 *	0.024 *	0.068	<0.001 *
ART, *n* (%)	15 (45.45%)	134 (55.14%)	36 (54.55%)	0.516	0.237	0.278	0.880
Monochorionic, *n* (%)	9 (27.27%)	59 (24.18%)	10 (15.15%)	0.059	0.692	0.064	0.036 *
Dichorionic, *n* (%)	24 (72.73%)	185 (75.82%)	56 (84.85%)				
Chronic HTN, *n* (%)	0 (0.00%)	0 (0.00%)	2 (3.03%)	<0.001 *		0.372	0.001 *
Preeclampsia, *n* (%)	1 (3.03%)	32 (13.11%)	4 (6.06%)	0.007 *	0.030 *	0.566	0.037 *
Overt DM, *n* (%)	1 (3.03%)	0 (0.00%)	1 (1.52%)	0.003 *	0.006 *	0.858	0.063
Gestational DM, *n* (%)	3 (9.09%)	25 (10.25%)	18 (27.27%)	<0.001 *	0.941	0.006 *	<0.001 *
Placenta previa, *n* (%)	2 (6.06%)	7 (2.87%)	1 (1.52%)	0.027 *	0.014 *	0.101	0.264
Threatened preterm, *n* (%)	27 (81.82%)	142 (58.20%)	38 (57.58%)	0.001 *	<0.001 *	0.001 *	0.977
PROM, *n* (%)	10 (30.30%)	59 (24.18%)	17 (25.76%)	0.549	0.354	0.612	0.795
Short cervix, *n* (%)	17 (51.52%)	118 (48.36%)	26 (39.39%)	0.138	0.726	0.142	0.083
Cause of delivery, *n* (%)				0.081	0.035 *	0.038 *	0.531
Elective	7 (21.21%)	89 (36.63%)	26 (39.39%)				
Spontaneous	20 (60.61%)	111 (45.68%)	31 (46.97%)				
Iatrogenic	6 (18.18%)	83 (17.70%)	9 (13.64%)				

* *p* value < 0.05. Underweight, BMI < 18.5; normal, 18.5 < BMI < 25; overweight, BMI > 25; *p* value α, between underweight BMI and normal BMI; *p* value β, between underweight BMI and overweight BMI; *p* value γ, between normal BMI and overweight BMI; BMI, body mass index; ART, artificial reproductive technique; HTN, hypertension; DM, diabetes mellitus; PROM, premature rupture of amniotic membrane.

**Table 2 medicina-58-01209-t002:** The neonatal outcomes according to maternal pre-pregnant BMI in twin pregnancy.

	Under Weight (*n* = 66)	Normal Weight (*n* = 488)	Over Weight (*n* = 132)	*p* Value	*p* Value α	*p* Value β	*p* Value γ
Gestational age at delivery, weeks	34.08 ± 2.90	34.93 ± 2.89	34.82 ± 2.66	0.220	0.025 *	0.075	0.696
Sex (male), *n* (%)	30 (45.45%)	259 (53.07%)	62 (46.97%)	0.286	0.302	0.960	0.251
Birthweight, gram	1939.02 ± 228.46	2193.02 ± 521.92	2244.05 ± 549.78	0.001 *	<0.001 *	<0.001 *	0.325
Birthweight percentile	27.36 ± 20.95	30.14 ± 21.23	36.61 ± 22.46	0.001 *	0.319	0.006 *	0.002 *
Placental weight, gram	958.79 ± 228.46	1029.94 ± 206.78	1107.42 ± 252.54	<0.001 *	0.010 *	<0.001 *	0.001 *
SGA, *n* (%)	13 (19.70%)	93 (19.06%)	17 (12.88%)	0.240	1.000	0.293	0.128
AGA–AGA, *n* (%)	42 (63.64%)	326 (66.80%)	100 (75.56%)	0.160	0.593	0.193	0.072
AGA–SGA, *n* (%)	22 (33.33%)	138 (28.28%)	30 (22.73%)				
SGA–SGA, *n* (%)	2 (3.03%)	24 (4.92%)	2 (1.52%)				
Twin weight discordance, *n* (%)	14 (21.21%)	96 (19.67%)	32 (24.24%)	0.513	0.897	0.766	0.303
Apgar score at 1 min (<7), *n* (%)	23 (34.85%)	116 (23.77%)	34 (25.76%)	0.149	0.072	0.244	0.720
Apgar score at 5 min (<7), *n* (%)	1 (1.52%)	18 (3.69%)	7 (5.30%)	0.411	0.582	0.372	0.557
NICU admission, (%)	52 (78.79%)	341 (70.16%)	94 (71.21%)	0.636	0.321	0.332	0.847
Oxygen supply, *n* (%)	33 (50.00%)	222 (45.68%)	67 (51.15%)	0.478	0.597	0.999	0.311
Intubation, *n* (%)	17 (25.76%)	75 (15.53%)	17 (12.98%)	0.060	0.056	0.041 *	0.557
RDS, *n* (%)	15 (22.73%)	64 (13.17%)	12 (9.16%)	0.030 *	0.058	0.017 *	0.276

* *p* value < 0.05. Underweight, BMI < 18.5; normal, 18.5 < BMI < 25; overweight, BMI > 25; *p* value α, between underweight BMI and normal BMI; *p* value β, between underweight BMI and overweight BMI; *p* value γ, between normal BMI and overweight BMI. SGA, small for gestational age; AGA, adequate for gestational age; NICU, neonatal intensive care unit; RDS.

**Table 3 medicina-58-01209-t003:** Comparisons on obstetrical complications, neonatal prognosis, and maternal weight gain between the AGA–AGA group and AGA–SGA or SGA–SGA group, regardless of BMI using maternal pre-pregnancy weight in twin pregnancy.

	AGA–AGA (*n* = 234)	AGA–SGA or SGA–SGA (*n* = 109)	*p* Value
Age, years	31.67 ± 4.27	33.67 ± 3.97	<0.001 *
Nulliparous, *n* (%)	153 (65.38%)	83 (76.15%)	0.001 *
ART, *n* (%)	114 (48.93%)	71 (65.14%)	<0.001 *
Monochorionic, *n* (%)	54 (23.08%)	24 (22.02%)	0.834
Dichorionic, *n* (%)	180 (76.92%)	85 (77.98%)	
Preeclampsia, *n* (%)	19 (8.12%)	18 (16.51%)	0.002 *
Gestational DM, *n* (%)	32 (13.68%)	14 (12.84%)	0.859
50 g GTT	131.83 ± 30.88	131.93 ± 29.28	0.976
TSH	2.04 ± 1.59	2.52 ± 2.28	0.007 *
Placenta previa, *n* (%)	10 (2.14%)	5 (4.59%)	0.074
Threatened preterm labor, *n* (%)	162 (69.23%)	45 (41.28%)	<0.001 *
PROM, *n* (%)	68 (29.06%)	18 (16.51%)	0.001 *
Short cervix, *n* (%)	126 (53.85%)	28 (29.47%)	<0.001 *
Postpartum bleeding, *n* (%)	15 (6.41%)	6 (5.50%)	0.772
Cause of delivery, *n* (%)			<0.001 *
Elective	76 (32.48%)	46 (42.59%)	
Spontaneous	129 (55.13%)	33 (30.56%)	
Iatrogenic	29 (12.39%)	29 (26.85%)	
Gestational age at delivery, weeks	34.37 ± 2.99	35.81 ± 2.24	<0.001 *
Gender, male, *n* (%)	244 (47.86%)	107 (49.08%)	0.507
Birthweight, grams	77.06	2117.34 ± 474.37	0.030 *
Birthweight percentile, %	38.40 ± 19.41	15.47 ± 17.36	<0.001 *
Placental weight, gram	1057.80 ± 228.26	995.50 ± 200.74	<0.001 *
Twin weight discordance, *n* (%)	42 (8.97%)	100 (45.87%)	<0.001 *
Apgar score at 1 min (<7), *n* (%)	126 (26.92%)	47 (21.56%)	0.158
Apgar score at 5 min (<7), *n* (%)	17 (3.63%)	9 (4.13%)	0.919
NICU admission, *n* (%)	319 (68.45%)	168 (77.06%)	0.025 *
Maternal height, kg	161.40 ± 5.10	159.85 ± 4.72	<0.001 *
Pre-pregnancy weight, kg	58.52 ± 11.07	55.83 ± 9.41	0.001 *
Weight at 15–18 weeks, kg	61.25 ± 10.16	57.86 ± 8.61	<0.001 *
Weight at 20–24 weeks, kg	64.13 ± 9.75	61.87 ± 8.96	0.059
Weight at 24–28 weeks, kg	67.50 ± 10.21	64.11 ± 7.95	0.001 *
Weight at delivery, kg	72.26 ± 12.30	69.65 ± 9.15	0.002 *

* *p* value < 0.05. AGA, adequate for gestational age; SGA, small for gestational age; ART, artificial reproductive technique; DM, diabetes mellitus; GTT, glucose tolerance test; TSH, thyroid stimulating hormone; PROM, premature rupture of amniotic membrane; NICU, neonatal intensive care unit.

**Table 4 medicina-58-01209-t004:** Comparisons on maternal weight and weight gain according to gestational age between the AGA–AGA group and AGA–SGA or SGA–SGA group in normal BMI twin pregnancy.

	AGA–AGA (*n* = 162)	AGA–SGA or SGA–SGA (*n* = 79)	*p* Value
Gestational age at delivery (weeks)	34.40 ± 3.02	35.99 ± 2.28	<0.001 *
Birthweight (grams)	2214.83 ± 555.64	2151.27 ± 443.72	0.176
Birthweight percentile (%)	37.88 ± 19.40	14.72 ± 16.57	<0.001 *
Maternal height (cm)	161.32 ± 5.22	160.14 ± 4.25	0.008 *
Pre-pregnancy weight (kg)	55.10 ± 5.54	54.51 ± 5.21	0.263
Weight at 15–18 weeks	58.42 ± 6.37	56.19 ± 5.64	0.004 *
Weight at 20–24 weeks	62.75 ± 6.91	61.23 ± 5.51	0.091
Weight at 24–28 weeks	65.54 ± 6.93	63.27 ± 5.23	0.003 *
Weight at delivery (kg)	69.49 ± 8.38	69.12 ± 5.95	0.578
Weight gain during the pregnancy (kg)	14.39 ± 5.88	14.61 ± 4.76	0.658
Weight gain during the 1st trimester (kg)	2.89 ± 3.49	2.47 ± 2.90	0.289
Weight gain during the 2nd trimester (kg)	7.21 ± 2.87	6.50 ± 2.04	0.043 *
Weight gain during the 3rd trimester (kg)	5.00 ± 3.67	6.27 ± 2.96	0.003 *

* *p* value < 0.05. AGA, adequate for gestational age; SGA, small for gestational age.

**Table 5 medicina-58-01209-t005:** Comparisons on maternal weight and weight gain according to gestational age between the AGA–AGA group and AGA–SGA or SGA–SGA group in overweight BMI twin pregnancy.

	AGA–AGA (*n* = 49)	AGA–SGA or SGA–SGA (*n* = 16)	*p* Value
Gestational age at delivery (weeks)	34.54 ± 2.82	35.52 ± 1.97	0.032 *
Birthweight (grams)	2269.13 ± 565.92	2140.94 ± 498.56	0.255
Birthweight percentile (%)	41.64 ± 20.16	21.81 ± 23.28	<0.001 *
Maternal height (cm)	160.97 ± 5.09	158.38 ± 6.25	0.020 *
Pre-pregnancy weight (kg)	74.42 ± 8.95	70.62 ± 10.90	0.051
Weight at 15–18 weeks	75.84 ± 8.49	70.43 ± 5.79	0.001 *
Weight at 20–24 weeks	77.43 ± 9.26	74.83 ± 6.19	0.315
Weight at 24–28 weeks	82.85 ± 9.85	75.61 ± 6.07	0.001 *
Weight at delivery (kg)	86.42 ± 11.06	80.29 ± 11.77	0.008 *
Weight gain during the pregnancy (kg)	12.01 ± 5.77	9.67 ± 4.02	0.013 *
Weight gain during the 1st trimester (kg)	2.92 ± 2.99	2.27 ± 2.69	0.354
Weight gain during the 2nd trimester (kg)	4.40 ± 2.02	4.57 ± 1.55	0.751
Weight gain during the 3rd trimester (kg)	3.73 ± 3.04	3.96 ± 2.09	0.765

* *p* value < 0.05. AGA, adequate for gestational age; SGA, small for gestational age.

**Table 6 medicina-58-01209-t006:** Comparisons on maternal weight and weight gain according to gestational age between the AGA–AGA group and AGA–SGA or SGA–SGA group in underweight BMI twin pregnancy.

	AGA–AGA (*n* = 42)	AGA–SGA or SGA–SGA (*n* = 24)	*p* Value
Gestational age at delivery (weeks)	33.49 ± 3.09	35.11 ± 2.24	0.028 *
Birthweight (grams)	1963.21 ± 519.92	1896.67 ± 580.37	0.633
Birthweight percentile (%)	35.83 ± 20.34	12.54 ± 11.94	<0.001 *
Maternal height (cm)	162.48 ± 3.82	159.67 ± 5.30	0.015 *
Pre-pregnancy weight (kg)	45.81 ± 3.54	44.25 ± 4.72	0.133
Weight at 15–18 weeks	50.17 ± 4.03	46.78 ± 4.35	0.022 *
Weight at 20–24 weeks	51.06 ± 3.11	50.00 ± 4.76	0.470
Weight at 24–28 weeks	56.33 ± 4.61	54.30 ± 5.01	0.215
Weight at delivery (kg)	58.59 ± 6.88	57.74 ± 5.88	0.616
Weight gain during the pregnancy (kg)	12.78 ± 5.02	13.49 ± 5.97	0.605
Weight gain during the 1st trimester (kg)	2.99 ± 2.38	3.21 ± 2.76	0.804
Weight gain during the 2nd trimester (kg)	6.57 ± 1.65	6.33 ± 3.45	0.830
Weight gain during the 3rd trimester (kg)	4.49 ± 4.10	3.69 ± 2.89	0.512

* *p* value < 0.05. AGA, adequate for gestational age; SGA, small for gestational age.

## Data Availability

The datasets used and/or analyzed during the current study are available from the corresponding author upon reasonable request.

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
