# Peer review of "Correlation between Maternal Weight Gain in Each Trimester and Fetal Growth According to Pre-Pregnancy Maternal Body Mass Index in Twin Pregnancies"

_medicina, 2022, doi:10.3390/medicina58091209_

Round 1

Reviewer 1 Report

Monitoring the weight gain of the pregnant woman can provide important information related to fetal growth, being a topic with wide clinical applicability. The topic is relevant and exciting to the field of the journal. The article makes a significant contribution to the field. The text is clear and easy to read. The manuscript has an excellent methodical description. The overall paper is organized and well written. The methods, the overall study design, and statistical analysis are clearly described. Discussions Section presents other similar research findings. The literature reviews are insightful and informative. The tables are well presented and easy to read and understand. The presented aspects sufficiently support the conclusions.

I have only a remark to make: at Line156 it is used the ROM abbreviation for the first time. It has no explanation. Please verify if you want to write PROM or ROM.

Author Response

Thank you for your precious comments. 

Reviewer's comment : at Line156 it is used the ROM abbreviation for the first time. It has no explanation. Please verify if you want to write PROM or ROM.

Answer : I corrected ROM to rupture of membrane (ROM) at line 156. And it includes all the membrane rupture such as PROM(premature rupture of membbrane) or PPROM (preterm premature rupture of membrane).

Reviewer 2 Report

The authors presented a retrospective study to determine the correlation between maternal weight gain in each trimester and fetal growth according to pre-pregnancy maternal body mass index in twin pregnancies. 

MAJOR REMARKS:

1.     The article needs to be reviewed by a native-speaker.

2.     It is customary to provide an Enhancing the QUAlity and Transparency Of health Research (EQUATOR) checklist depending on the type of study. This may not be required by the journal, but it is good practice.

3.     Some confounding factors that may have influenced fetal growth, regardless of maternal weight gain (i.e., differences in fetal flowmetry), need to be considered.

4.     The modality in which the study population size and the number of participants for each BMI group were calculated should be clarified.

5.     The availability of the datasets analyzed is encouraged to further increase the transparency of your research.

MINOR REMARKS:

p. 2, line 75: or → instead
